# Spatial Correspondence between Graph Neural Network-Segmented Images

**Qian Li** [1,2]                                      LIQIAN96@HIT.EDU.CN

[1] *State Key Laboratory of Robotics and System, Harbin Institute of Technology, Harbin, China*
[2] *Department of Medical Physics and Biomedical Engineering, University College London, London, U.K*

**Yunguan Fu**[2]                                   YUNGUAN.FU.18@UCL.AC.UK
**Qianye Yang**[2]                                QIANYE.YANG.19@UCL.AC.UK
**Zhijiang Du**[1]                                    DUZJ01@HIT.EDU.CN
**Hongjian Yu**[1]                                   YUHONGJIAN@HIT.EDU.CN
**Yipeng Hu**[2]                                     YIPENG.HU@UCL.AC.UK

**Editors:** Accepted for publication at MIDL 2023

## Abstract

Graph neural networks (GNNs) have been proposed for medical image segmentation, by predicting anatomical structures represented by graphs of vertices and edges. One such type of graph is predefined with fixed size and connectivity to represent a reference of anatomical regions of interest, thus known as templates. This work explores the potentials in these GNNs with common topology for establishing spatial correspondence, implicitly maintained during segmenting two or more images. With an example application of registering local vertebral sub-regions found in CT images, our experimental results showed that the GNN-based segmentation is capable of accurate and reliable localization of the same interventionally interesting structures between images, not limited to the segmentation classes. The reported average target registration errors of 2.2±1.3 mm and 2.7±1.4 mm, for aligning holdout test images with a reference and for aligning two test images, respectively, were by a considerable margin lower than those from the tested non-learning and learning-based registration algorithms. Further ablation studies assess the contributions towards the registration performance, from individual components in the originally segmentation-purposed network and its training algorithm. The results highlight that the proposed segmentation-in-lieu-of-registration approach shares methodological similarities with existing registration methods, such as the use of displacement smoothness constraint and point distance minimization albeit on non-grid graphs, which interestingly yielded benefits for both segmentation and registration. We, therefore, conclude that the template-based GNN segmentation can effectively establish spatial correspondence in our application, without any other dedicated registration algorithms.

**Keywords:** Registration, graph neural networks, orthopedic surgery.

## 1. Introduction

Graph neural networks (GNNs) provide versatility in representing data sampled from non-grid spatial locations using connected vertices and edges. For medical imaging applications, GNNs have been proposed to represent the input images and extract features for tasks such as classification and registration (Sun et al., 2021), as well as used in decoding for segmentation tasks (Han et al., 2022; Fu et al., 2021), and representing non-grid prediction output for

segmentation. In the latter, graph templates are designed for the regions of interest (ROIs) to segment. For example, (Wickramasinghe et al., 2020) deforms a spherical mesh template to segment the liver. (Kong et al., 2021) and (Kong and Shadden, 2021) used four templates to describe the four parts of the heart. In (Bongratz et al., 2022) and (Hoopes et al., 2021), a smoothed cortex model is deformed to segment the cerebral cortex.

In these studies, networks were trained to deform the predefined template meshes iteratively to fit the object surface in the input image to achieve mesh reconstruction or segmentation. We observed that the correspondence, defined by the same vertices before and after mesh deformation, pertains anatomically corresponding locations, but was understandably discarded for segmentation tasks. In this paper, this correspondence is used to register the input image with the predefined template mesh (we call it a reference mesh) and further register the input image pairs.

To demonstrate the application of the proposed registration strategy, we take annotating spinal vertebrae from CT images as an example, which was previously achieved with convolutional neural networks (CNNs) such as variants of UNet (Li et al., 2021; Lessmann et al., 2019). While localizing finer vertebral sub-regions is also desirable in a number of surgical tasks, and recent robot-assisted surgery may also benefit from precise planning of robotic trajectories, with respect to these local anatomies (Hu et al., 2013; Dillon et al., 2016). Atlas registration can be considered a suitable method in the absence of a sufficient number of labeled data sets. It also has the potential to transfer the planned surgical trajectories from the atlas to new images.

In this work, we first validate both classical intensity-based and recent learning-based registration algorithms. Moreover, we propose template-based GNNs to represent vertebra segmentation output and infer the spatial correspondence from the segmented vertebrae, a denser, more local correspondence between sub-regions without supervision other than the corresponding segmentation classes (the entire vertebra versus background in this case). This is enabled by the spatial connectivity from the GNNs, inherent within the common template. Interestingly, the experiments show that graph-segmentation-derived dense correspondence achieved significantly lower target registration errors (TREs), compared with the tested registration algorithms.

The contributions of this paper can be summarized as follows.

- A previous segmentation network (Bongratz et al., 2022) was reused for image registration tasks.

- Based on predefined reference meshes, strategies for a reference to target registration and a general pairwise image registration are proposed.

- The proposed method achieves significantly better performance on both target point localization and atlas segmentation tasks, compared with the tested classical non-learning and other learning-based registration algorithms.

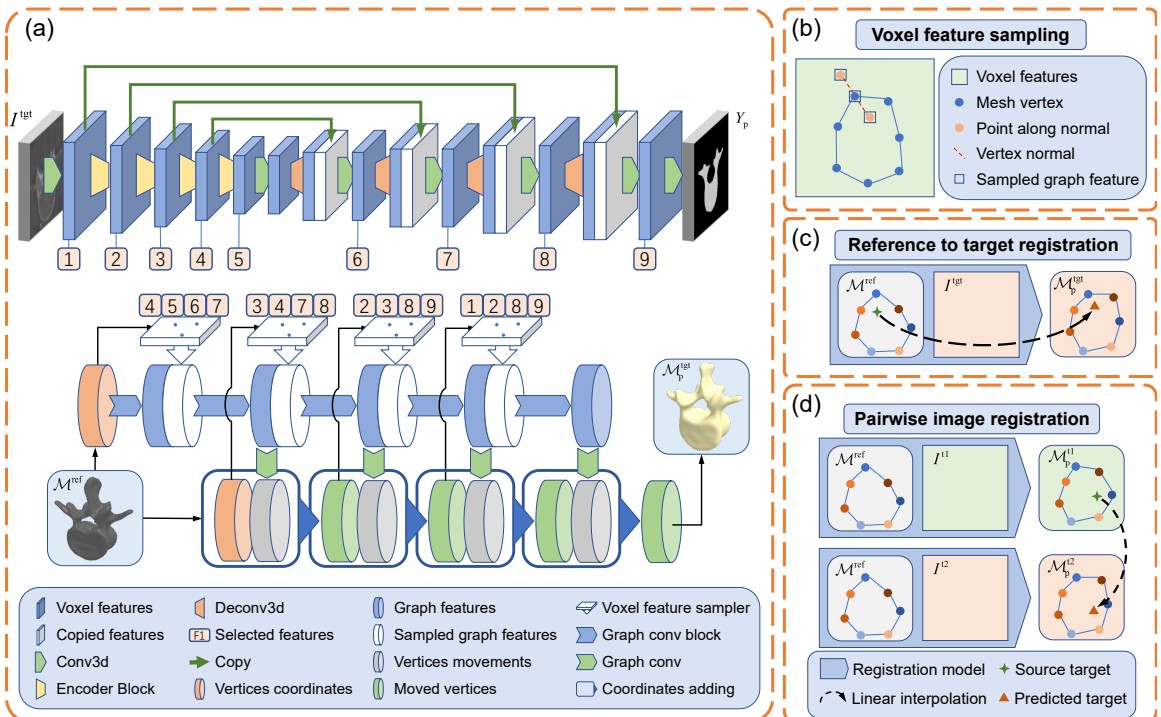

Figure 1: (a) Overview of the registration network with a GNN module and a CNN module. (b) The illustration of feature sampling. (c) Reference to target registration. (d) Pairwise image registration. Symbols are defined in the text.

## 2. Registration with a Reference Mesh

In this section, we provide the details of the proposed approach based on CNN and GNN, to register a reference mesh (i.e. the template in the context of segmentation used in previous studies), from a reference image to the given voxel image.

The proposed registration method aims to register a set of predefined surface points in the reference image $I^{\text{ref}}$ with those in the target image $I^{\text{tgt}}$. A smoothed surface mesh from the training data sets is used as the reference mesh, which can be represented by sets of vertices, edges, and faces, i.e. $\mathcal{M}^{\text{ref}} = (\mathcal{V}^{\text{ref}}, \mathcal{E}^{\text{ref}}, \mathcal{F}^{\text{ref}})$. The registration task is to predict the displacements $\mathcal{D}_{\text{p}} = f_\theta(I^{\text{tgt}}, \mathcal{M}^{\text{ref}})$ from the input image and reference mesh, where $f_\theta$ is a neural network with parameters $\theta$. Applying the displacement to $\mathcal{M}^{\text{ref}}$ results in a deformed mesh $\mathcal{M}_{\text{p}}^{\text{tgt}} = (\mathcal{V}_{\text{p}}^{\text{tgt}}, \mathcal{E}_{\text{p}}^{\text{tgt}}, \mathcal{F}_{\text{p}}^{\text{tgt}})$ for the $I^{\text{tgt}}$. That is, for any vertex $\boldsymbol{v}^{\text{ref}} \in \mathcal{V}^{\text{ref}}$, let the displacement be $\boldsymbol{d}_{\text{p}}$, the moved vertex $\boldsymbol{v}_{\text{p}}^{\text{tgt}}$ is calculated by $\boldsymbol{v}^{\text{ref}} + \boldsymbol{d}_{\text{p}}$. Therefore, a series of corresponding points $\mathcal{R}_{\text{p}}^{\text{ref,tgt}} = \{(\boldsymbol{v}^{\text{ref}}, \boldsymbol{v}_{\text{p}}^{\text{tgt}}) \mid \boldsymbol{v}^{\text{ref}} \in \mathcal{V}^{\text{ref}}, \boldsymbol{v}_{\text{p}}^{\text{tgt}} \in \mathcal{V}_{\text{p}}^{\text{tgt}}\}$ from the reference image to the target image are generated through the proposed registration method. For a new target point $\boldsymbol{p}^{\text{ref}}$ in $I^{\text{ref}}$, the registered corresponding point in $I^{\text{tgt}}$ for it can be obtained by using the piecewise linear interpolator $\boldsymbol{p}_{\text{p}}^{\text{tgt}} = \Phi(\mathcal{R}_{\text{p}}^{\text{ref,tgt}}, \boldsymbol{p}^{\text{ref}})$, where $\Phi(\mathcal{R}^{\text{a,b}}, \boldsymbol{p}^a)$ denotes the interpolated coordinate at $\boldsymbol{p}^a$ using a series of paired points in a and b. Figure 1 illustrates this reference-to-target registration process.

More generally, the pairwise registration method registers the set of predefined surface points from one image $I^{\text{t1}}$ to a second image $I^{\text{t2}}$, illustrated in Figure 1. Denote the corre-

sponding deformed meshes as $\mathcal{M}_{\mathrm{p}}^{\mathrm{t}1} = (\mathcal{V}_{\mathrm{p}}^{\mathrm{t}1}, \mathcal{E}_{\mathrm{p}}^{\mathrm{t}1}, \mathcal{F}_{\mathrm{p}}^{\mathrm{t}1})$ and $\mathcal{M}_{\mathrm{p}}^{\mathrm{t}2} = (\mathcal{V}_{\mathrm{p}}^{\mathrm{t}2}, \mathcal{E}_{\mathrm{p}}^{\mathrm{t}2}, \mathcal{F}_{\mathrm{p}}^{\mathrm{t}2})$ respectively. With the vertex displacement from the input to the target, the proposed registration method can be applied by registering the reference mesh $\mathcal{M}^{\mathrm{ref}}$ to $I^{\mathrm{t}1}$ and $I^{\mathrm{t}2}$ separately, with displacements $\mathcal{D}^{\mathrm{t}1} = f_\theta(I^{\mathrm{t}1}, \mathcal{M}^{\mathrm{ref}})$ and $\mathcal{D}^{\mathrm{t}2} = f_\theta(I^{\mathrm{t}2}, \mathcal{M}^{\mathrm{ref}})$ respectively. For any vertex $\boldsymbol{v}^{\mathrm{ref}} \in \mathcal{V}^{\mathrm{ref}}$, let the moved vertices in the two images be $\boldsymbol{v}^{\mathrm{ref}} + \boldsymbol{d}_{\mathrm{p}}^{\mathrm{t}1}$ and $\boldsymbol{v}^{\mathrm{ref}} + \boldsymbol{d}_{\mathrm{p}}^{\mathrm{t}2}$. The relative displacement between the two images for $\boldsymbol{v}^{\mathrm{ref}}$ is $\boldsymbol{d}_{\mathrm{p}}^{\mathrm{t}2} - \boldsymbol{d}_{\mathrm{p}}^{\mathrm{t}1}$ and the correspondence between the two moved vertices is established as $\boldsymbol{v}_{\mathrm{p}}^{\mathrm{t}2} = \boldsymbol{v}_{\mathrm{p}}^{\mathrm{t}1} - \boldsymbol{d}_{\mathrm{p}}^{\mathrm{t}1} + \boldsymbol{d}_{\mathrm{p}}^{\mathrm{t}2}$. Therefore, in the pairwise registration task, for a point $\boldsymbol{p}^{\mathrm{t}1}$ from $I^{\mathrm{t}1}$, the corresponding point in $I^{\mathrm{t}2}$ can be predicted as $\boldsymbol{p}_{\mathrm{p}}^{\mathrm{t}2} = \Phi(\mathcal{R}_{\mathrm{p}}^{\mathrm{t}1,\mathrm{t}2}, \boldsymbol{p}^{\mathrm{t}1})$, where $\mathcal{R}_{\mathrm{p}}^{\mathrm{t}1,\mathrm{t}2} = \{(\boldsymbol{v}_{\mathrm{p}}^{\mathrm{t}1}, \boldsymbol{v}_{\mathrm{p}}^{\mathrm{t}2}) | \boldsymbol{v}_{\mathrm{p}}^{\mathrm{t}1} \in \mathcal{V}_{\mathrm{p}}^{\mathrm{t}1}, \boldsymbol{v}_{\mathrm{p}}^{\mathrm{t}2} \in \mathcal{V}_{\mathrm{p}}^{\mathrm{t}2}\}$.

## 3. Network Construction and the Training Loss

In this work, the neural network with both CNN and GNN modules from Bongratz et al. (2022) is adopted, illustrated in Figure 1. The U-Net-like CNN module ingests the input image and predicts a segmentation mask for the vertebra. The GNN module takes the reference mesh as input and performs graph convolution with vertex features extracted from the CNN module to adjust vertex coordinates progressively. Formally, at each graph convolution layer, denote the vertex $\boldsymbol{v}_i$'s features in GNN as $\boldsymbol{f}_{i,\mathrm{GNN}}$, it is updated by aggregating vertex features of neighbors and itself from both GNN and CNN modules:

$$\boldsymbol{f}_{i,\mathrm{GNN}} = h\left(\frac{1}{1 + |\mathcal{N}(i)|}\left(W_0 \boldsymbol{f}_i + \boldsymbol{b}_0 + \sum_{j \in \mathcal{N}(i)} (W_1 \boldsymbol{f}_j + \boldsymbol{b}_1)\right)\right) \tag{1}$$

$$\boldsymbol{f}_i = \mathrm{concat}[\boldsymbol{f}_{i,\mathrm{CNN}}, \hat{\boldsymbol{f}}_{i,\mathrm{GNN}}] \tag{2}$$

where $h$ is Relu activation layer; $W_0, \boldsymbol{b}_0, W_1, \boldsymbol{b}_1$ are learnable weights; $\mathcal{N}(i)$ is the neighbour vertices of $\boldsymbol{v}_i$ ; and $\boldsymbol{f}_{i,\mathrm{CNN}}$ is the features extracted from CNN features and $\hat{\boldsymbol{f}}_{i,\mathrm{GNN}}$ is previous graph features. $\boldsymbol{f}_{i,\mathrm{CNN}}$ is calculated by concatenating the sampled CNN embeddings $H$ at multiple points along the vertex normal vector $\boldsymbol{n}_i$:

$$\boldsymbol{f}_{i,\mathrm{CNN}} = \mathrm{concat}_{\alpha_k \in \alpha}[\phi(H, \boldsymbol{v}_i + \alpha_k \boldsymbol{n}_i)], \tag{3}$$

with $\phi(H, \boldsymbol{v})$ representing the sampled embedding from CNN embedding $H$ at point $\boldsymbol{v}$ and $\alpha$ is the predefined distances list. Such sampling is expected to provide more context beyond the surface and facilitate the model training, described as follows.

To train the network $f_\theta$, a composed loss functions Bongratz et al. (2022) is adapted:

$$\begin{aligned} \mathcal{L} = \ &\lambda_{\mathrm{seg}}(t)\mathcal{L}_{\mathrm{seg}}(Y_{\mathrm{p}}, Y_{\mathrm{gt}}) + \lambda_{\mathrm{delay}}(t)(\lambda_{\mathrm{chamfer}}\mathcal{L}_{\mathrm{chamfer}}(\mathcal{M}_{\mathrm{p}}^{\mathrm{tgt}}, \mathcal{M}_{\mathrm{gt}}^{\mathrm{tgt}}) \\ &+ \lambda_{\mathrm{norm,inter}}\mathcal{L}_{\mathrm{norm,inter}}(\mathcal{M}_{\mathrm{p}}^{\mathrm{tgt}}, \mathcal{M}_{\mathrm{gt}}^{\mathrm{tgt}}) + \lambda_{\mathrm{norm,intra}}\mathcal{L}_{\mathrm{norm,intra}}(\mathcal{M}_{\mathrm{p}}^{\mathrm{tgt}}) \\ &+ \lambda_{\mathrm{edge}}(t)\mathcal{L}_{\mathrm{edge}}(\mathcal{M}_{\mathrm{p}}^{\mathrm{tgt}}) + \lambda_{\mathrm{disp}}\mathcal{L}_{\mathrm{disp}}(\mathcal{D}_{\mathrm{p}}, \mathcal{M}_{\mathrm{p}}^{\mathrm{tgt}})) \end{aligned} \tag{4}$$

where, $\mathcal{L}_{\mathrm{seg}}(Y_{\mathrm{p}}, Y_{\mathrm{gt}})$ is the binary cross entropy between the predicted and ground truth vertebral segmentation masks; $\mathcal{L}_{\mathrm{chamfer}}(\mathcal{M}_{\mathrm{p}}^{\mathrm{tgt}}, \mathcal{M}_{\mathrm{gt}}^{\mathrm{tgt}})$ is the curvature-weighted Chamfer loss that penalizes mismatched vertex positions between the predicted and ground truth meshes; $\mathcal{L}_{\mathrm{norm,inter}}(\mathcal{M}_{\mathrm{p}}^{\mathrm{tgt}}, \mathcal{M}_{\mathrm{gt}}^{\mathrm{tgt}})$ is the normal distance loss that penalizes mismatched vertex normal

vectors between the predicted and ground truth meshes; $\mathcal{L}_{\mathrm{norm,intra}}(\mathcal{M}_{\mathrm{p}}^{\mathrm{tgt}})$ is the normal distance loss that promotes the consistency of adjacent face normal vectors in the predicted mesh; $\mathcal{L}_{\mathrm{edge}}(\mathcal{M}_{\mathrm{p}}^{\mathrm{tgt}})$ is the edge length loss that penalizes long edges of the predicted mesh; and $\mathcal{L}_{\mathrm{disp}}(\mathcal{D}_{\mathrm{p}}, \mathcal{M}_{\mathrm{p}}^{\mathrm{tgt}})$ is the displacement regularisation loss which calculates the L2 norm of the predicted vertices displacements. Different from the Laplacian smoothing in Bongratz et al. (2022), we weighted the vertex displacement by the inverse of the edge length to account for differences in neighbors at different distances. The definitions of these loss functions are detailed in Appendix.

To avoid divergence at the initial training stage, frequently found in our preliminary experiments, a delayed weight strategy was adopted $\lambda_{\mathrm{delay}}(t) = 0.5 + \arctan[(t - 3000)]/\pi$, controlled by the number of steps $t$. A dynamic loss weighting mechanism is also empirically designed, $\lambda_{\mathrm{seg}}(t) = \lambda_{\mathrm{edge}}(t) = 0.5 - \arctan[(t - 10000)/1000]/\pi$.

## 4. Experiments and Results

### 4.1. Data sets and preprocessing

Three online published spine CT image segmentation data sets were used for training and testing, namely Lumbar vertebra segmentation CT image data sets (LumSeg), Spine and Vertebrae Segmentation Datasets (SpiSeg), and xVertSeg data sets (xVertSeg). The mixed data contain a total of 35 subjects with 175 lumbar vertebrae, and for each case, the original CT image and the voxel-labeled masks are given. The data were randomly divided into a training set of 24 subjects and a holdout test set of 11 subjects. All the results in the paper were based on the test set, without using a validation set for hyperparameter tuning, which may further improve the performance. All images and segmentation ground truth were resampled at a voxel dimension of $0.5mm \times 0.5mm \times 0.5mm$, which are randomly cropped, from the vertebral center, to a size of $128 \times 192 \times 192$ in advance.

The ground truth surface meshes were obtained by using the marching cube algorithm (Lorensen and Cline, 1987) based on the segmentation labels, followed by a Laplacian smoothing filter with the trimesh Python library. To validate sub-region registration, all vertices of the mesh were divided into 10 categories based on 9 boundary planes selected manually. An example of the selected planes is shown in the Appendix. Examples of the labeled mesh can be found in the GT of 3($a$) and the symbols represent spinous process (SP), left lamina (LL), right lamina (RL), left articular process (LAP), right articular process (RAP), left transverse process (LTP), right transverse process (RTP), left pedicle (LP), right pedicle (RP) and vertebral body (VB). We selected a registration target point for each category of the test data set mesh by averaging the coordinates of all vertices of this category.

An image from the training data sets was randomly selected as the predefined reference and the surface mesh was extracted with the marching cubes algorithm. The Laplacian smoothing algorithm was applied to the reference mesh to remove the unsmooth and personalized details. The experimental results were compared with three image registration baselines, the iterative intensity-based method NiftyReg (Modat et al., 2010), the learning-based method WeaklySup (Hu et al., 2018a,b; Fu et al., 2020) and VoxelMorph (Balakrishnan et al., 2019). NiftyReg was implemented with SSD as the similarity measure regularised by bending energy, with otherwise default configurations. Segmentation BCE loss was used to train WeaklySup and the combined loss of BCE and the similarity loss SSD was adopted for training VoxelMorph.

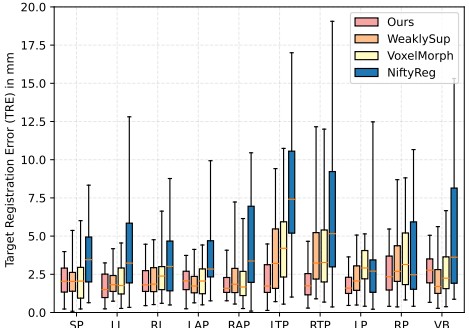
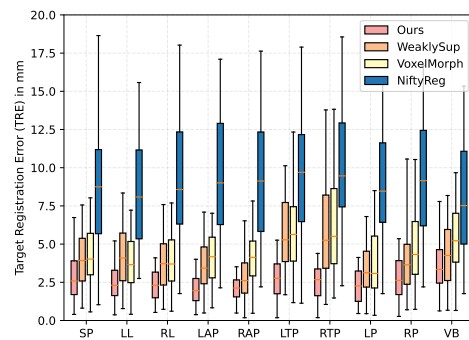

($a$) Reference to target registration.  ($b$) Arbitrary image pair registration.

Figure 2: The TREs results of Ours, WeaklySup, VoxelMorph, and NiftyReg. Detailed data can be found in the Appendix.

Both WeaklySup and VoxelMorph were trained in a weakly-supervised algorithm, with the same supervising labels, training set, and reference case. Results from a wider permutation of algorithms and loss functions are reported in Table 4 and Table 5 in the Appendix. All results are reported on the same test set. The official implementation of NiftyReg and VoxelMorph, and a PyTorch adaptation of WeaklySup (Yang et al., 2022) were used. All experiments were performed on NVIDIA GPU Quadro P5000.

## 4.2. Reference to target registration

The alignment between the fixed reference image and images in the test set was first quantified and the TREs based on the geometric centers of individual sub-regions are summarised in Figure 2($a$) and also in Table 3. Compared with the results from NiftyReg, the TREs are statistically lower for all sub-regions. The improvement over the learning-based algorithms was less evident, with lower TREs observed in eight out of ten sub-regions when compared with WeaklySup, indicating a comparable registration performance in this task. More registration results can be found in the Appendix.

## 4.3. Arbitrary image pair registration via reference

In this experiment, 100 vertebral pairs were randomly sampled from the test set, and the sub-region TREs are illustrated in Figure 2($b$). Our registration model achieved an average TRE on all sub-regions of 2.68±1.44 mm and outperformed WeaklySup (4.37±2.67 mm), VoxelMorph (4.79±2.68 mm) and NiftyReg (9.35±4.38 mm).

The choice of the predefined reference may be a source of bias, however, previous studies showed that segmentation tasks did not seem sensitive to such a smoothed mesh template (Bongratz et al., 2022). It was found to be a much stronger bias if the two test images are used as respective the reference and the target during test time - without using the intermediate predefined reference, as opposed to the two correspondence composing approaches described above. This indeed led to much higher TREs (9.44±15.36 mm). Results from the models trained with a variable reference (randomly sampled reference during training) are summarised in Section 4.5.

## 4.4. Vertebra and sub-region segmentation

For reference purposes, we also report the results based on segmentation metrics on both the sub-regions and the entire vertebra. Hausdorff distance (HD), the average symmetric surface distance (ASSD), and Dice score are summarised in Table 1 and Figure 3(a). More examples are provided in Appendix. Interestingly, our model achieves better results than the baselines. Some segmentation examples can be seen in Figure 3(b) and Table 2.

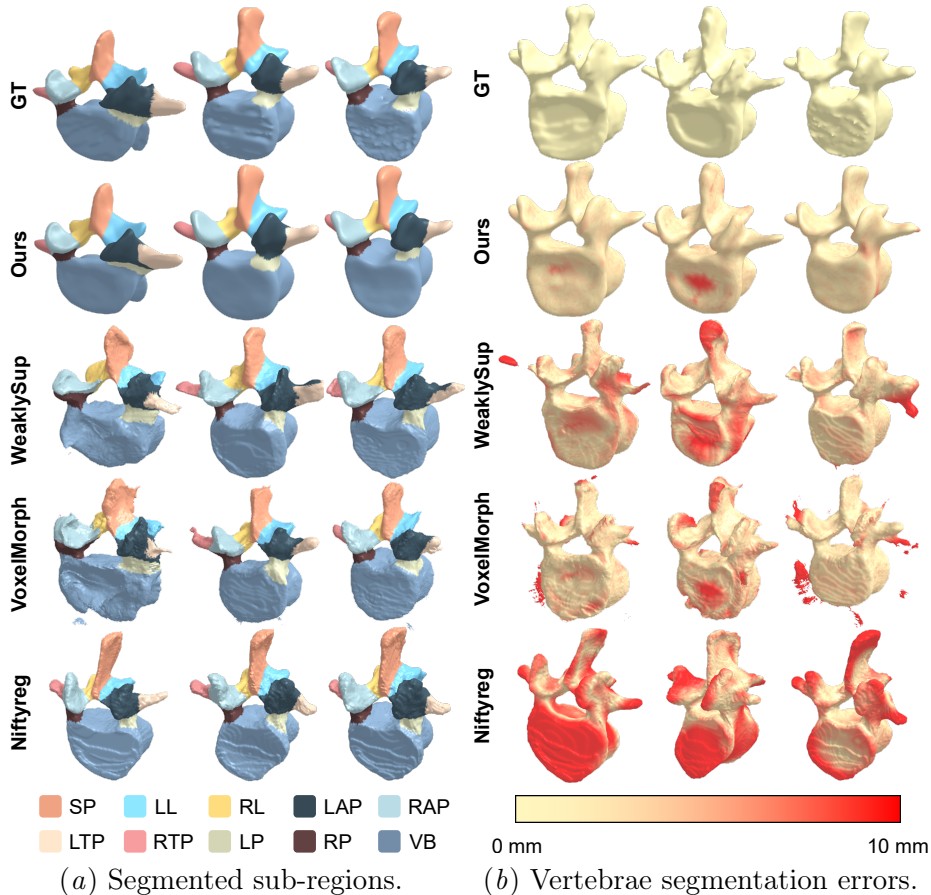

(a) Segmented sub-regions.  (b) Vertebrae segmentation errors.

Figure 3: Examples of sub-regions (a) and vertebra segmentation errors with a color error bar (b). Further examples are provided in Appendix.

## 4.5. Ablation studies

To better understand 1) the importance of network architecture and loss function design and 2) their respective contributions to both segmentation and registration tasks, we provide a set of ablation studies to compare the results when the following modification was independently made, summarised in Table 3. **Variable ref.**: This model was trained with a variable reference randomly from the training set, rather than a fixed reference and it was tested without using a fixed reference in pairwise registration experiments. **w/o norm. feat.**: Graph features were only interpolated from the voxel features by the mesh vertices which means $\alpha = [0]$ in Equation 3. **Constant $\lambda_{\mathbf{vox}}$**: The weight for voxel segmentation loss was

Table 1: Sub-regions segmentation HD results in mm with average±standard deviation, details also described in the text. Statistically significant results are in bold (paired t-tests with Bonferroni correction in the multiple comparisons at a significance level $\alpha$=0.001).

|  | SP | RL | LL | RTP | LTP |
|---|---|---|---|---|---|
| Ours | **0.97±1.29** | **0.78±0.56** | **0.84±0.55** | **1.06±1.23** | **1.44±2.27** |
| WeaklySup | 2.46±1.29 | 2.73±1.09 | 2.58±0.90 | 4.04±3.98 | 3.65±2.48 |
| VoxelMorph | 2.23±2.65 | 3.98±2.74 | 2.81±1.85 | 3.77±3.04 | 4.38±3.02 |
| NiftyReg | 4.98±2.92 | 5.59±3.69 | 5.67±4.24 | 6.78±4.78 | 8.46±4.51 |
|  | RAP | LAP | RP | LP | VB |
| Ours | **1.37±2.61** | **0.95±0.67** | 1.62±0.97 | 1.52±0.82 | **0.96±0.40** |
| WeaklySup | 3.12±2.66 | 2.54±0.99 | 2.77±1.42 | 2.57±1.36 | 3.32±2.34 |
| VoxelMorph | 3.08±2.93 | 2.15±1.03 | 2.27±2.00 | 1.87±1.46 | 3.46±2.49 |
| NiftyReg | 5.11±3.30 | 4.68±2.84 | 3.69±2.96 | 3.81±3.36 | 8.88±5.98 |

Table 2: Vertebra segmentation results.

| Models | HD (mm) | ASSD (mm) | Dice (%) |
|---|---|---|---|
| Ours | **1.14±0.49** | **0.54± 0.11** | **93.25± 1.47** |
| WeaklySup | 3.22±1.76 | 1.19± 0.46 | 86.50± 5.21 |
| VoxelMorph | 3.26± 1.97 | 0.98± 0.44 | 91.12±4.10 |
| NiftyReg | 7.97± 4.65 | 2.52± 1.48 | 71.82±15.50 |

set to a constant value during the training. **Classical chamfer**: Classical chamfer loss was used which is equal to set $\kappa(\cdot \mid \kappa_{\max}) = 1$ in Equation (6). **Laplacian**: Uniform weights were used when calculating the displacement regularisation loss which means $w(\boldsymbol{v}, \boldsymbol{v}_{\mathrm{nbr}}) = \frac{1}{\mathcal{N}(\boldsymbol{v})}$ in Equation (12) and it is equal to using the Laplacian smooth on the predicted displacements (Nealen et al., 2006). **w/o disp. reg.**: The model was trained without displacement regularisation loss. **Constant $\lambda_{\mathbf{edge}}$**: A constant weight for edge length loss was used when training.

Table 3: Ablation study results. TRE-reference and TRE-pair denote the performance from reference-to-target registration and pairwise registration experiments, described in Secs. 4.2 and 4.3, respectively. Other metrics are described in the text. The best results are in bold.

|  | TRE-reference | TRE-pair | HD | ASSD | Dice |
|---|---|---|---|---|---|
| Ours | **2.15±1.33** | **2.68±1.44** | 1.14±0.49 | 0.54±0.11 | 93.25±1.47 |
| Variable ref. | 5.48±2.81 | 5.48±5.39 | 3.16±0.84 | 0.98±0.20 | 87.22±3.75 |
| w/o norm. feat. | 2.42±1.51 | 3.09±1.60 | 1.40±0.63 | 0.61±0.15 | 93.10±2.25 |
| Constant $\lambda_{\mathrm{vox}}$ | 2.80±1.62 | 3.04±1.52 | 1.84±1.05 | 0.74±0.27 | 92.71±2.94 |
| Classical chamfer | 3.06±1.64 | 3.45±1.71 | 1.96±0.85 | 0.75±0.20 | 91.82±2.16 |
| Laplacian | 2.28±1.39 | 2.78±1.51 | **1.10±0.47** | **0.53±0.11** | 92.71±2.68 |
| w/o disp. reg. | 3.53±1.68 | 4.06±2.02 | 1.52±0.62 | 0.65±0.15 | 93.19±2.22 |
| Constant $\lambda_{\mathrm{edge}}$ | 3.03±1.49 | 3.22±1.58 | 1.41±0.83 | 0.61±0.20 | **93.81±2.19** |

## 5. Discussion

The proposed method uses GNN-represented meshes to describe object surfaces and predict vertex displacements between a reference mesh and one or more target images. Although the same structure as the previous network (Bongratz et al., 2022) is adopted, based on the vertices correspondence before and after the mesh deformation, the output result is used to establish the spatial correspondence between the reference mesh and the target image or between a pair of target images. This paper takes vertebral CT image registration as an example since the atlas registration can be applied in spinal surgery planning. It may be applicable in other atlas registration tasks, such as other orthopedic image registration or some soft tissue organ registration. However, those with unlabeled data sets driven only by intensity-based loss were not investigated in this work. As described in (Bongratz et al., 2022), the network is not guaranteed to be free of self-intersections. But probably because of the use of a structure specific reference mesh, they were not observed in the experiments.

## Acknowledgments

This work was supported by the Wellcome/EPSRC Centre for Interventional and Surgical Sciences [203145Z/16/Z] and the China Scholarship Council (CSC, No.202106120119).

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

## 6. Appendix

### 6.1. Loss Functions

#### 6.1.1. SEGMENTATION LOSS

The binary cross entropy between the predicted and ground truth vertebral segmentation masks $\mathcal{L}_{\text{seg}}(Y_{\text{p}}, Y_{\text{tgt}})$ is defined as:

$$\mathcal{L}_{\text{seg}}(Y_{\text{p}}, Y_{\text{tgt}}) = -\frac{1}{N} \sum_{i=1}^{N} (Y_{\text{gt},i} \log Y_{\text{p},i} + (1 - Y_{\text{gt},i}) \log(1 - Y_{\text{p},i})) \tag{5}$$

where $Y_{\text{gt}}$ is the binary mask and $Y_{\text{p}}$ is the predicted soft mask with values between $[0, 1]$. The subscript $i$ represents the value at voxel $i$ which iterates over all voxels in the image.

#### 6.1.2. CURVATURE-WEIGHTED CHAMFER LOSS

The curvature-weighted Chamfer loss that penalizes mismatched vertex positions between the predicted and ground truth meshes is defined as $\mathcal{L}_{\text{chamfer}}(\mathcal{M}_{\text{p}}^{\text{tgt}}, \mathcal{M}_{\text{gt}}^{\text{tgt}})$:

$$\begin{aligned}
\mathcal{L}_{\text{chamfer}}(\mathcal{M}_{\text{p}}^{\text{tgt}}, \mathcal{M}_{\text{gt}}^{\text{tgt}}) = {} & \frac{1}{\left|\mathcal{V}_{\text{gt}}^{\text{tgt}}\right|} \sum_{\boldsymbol{u} \in \mathcal{V}_{\text{gt}}^{\text{tgt}}} \kappa(\boldsymbol{u} \mid \kappa_{\max}) \min_{\mathbf{v} \in \mathcal{V}_{\text{p}}^{\text{tgt}}} \|\mathbf{u} - \mathbf{v}\|^2 \\
& + \frac{1}{\left|\mathcal{V}_{\text{p}}^{\text{tgt}}\right|} \sum_{\boldsymbol{v} \in \mathcal{V}_{\text{p}}^{\text{tgt}}} \kappa(\tilde{\boldsymbol{u}} \mid \kappa_{\max}) \min_{\mathbf{u} \in \mathcal{V}_{\text{gt}}^{\text{tgt}}} \|\mathbf{v} - \mathbf{u}\|^2,
\end{aligned} \tag{6}$$

with $\tilde{\boldsymbol{u}}$ being the closest vertex in $\mathcal{V}_{\text{gt}}^{\text{tgt}}$ to $\boldsymbol{v}$, i.e. $\tilde{\boldsymbol{u}} = \arg\min_{\mathbf{u} \in \mathcal{V}_{\text{gt}}^{\text{tgt}}} \|\mathbf{v} - \mathbf{u}\|^2$. $\kappa(\cdot \mid \kappa_{\max})$ is the curvature function defined in Bongratz et al. (2022) with discrete mean curvature function $\bar{\kappa}(\cdot)$ defined in Cohen-Steiner and Morvan (2003):

$$\kappa(\boldsymbol{v} \mid \kappa_{\max}) = \min(1 + \bar{\kappa}(\boldsymbol{v}), \kappa_{\max}). \tag{7}$$

#### 6.1.3. INTER-MESH NORMAL DISTANCE LOSS

The normal distance loss that penalizes mismatched vertex normal vectors between the predicted and ground truth meshes is defined as $\mathcal{L}_{\text{norm,inter}}(\mathcal{M}_{\text{p}}^{\text{tgt}}, \mathcal{M}_{\text{gt}}^{\text{tgt}})$:

$$\begin{aligned}
\mathcal{L}_{\text{norm,inter}}(\mathcal{M}_{\text{p}}^{\text{tgt}}, \mathcal{M}_{\text{gt}}^{\text{tgt}}) = {} & \frac{1}{\left|\mathcal{V}_{\text{gt}}^{\text{tgt}}\right|} \sum_{\boldsymbol{u} \in \mathcal{V}_{\text{gt}}^{\text{tgt}}} 1 - \cos(\boldsymbol{n}(\boldsymbol{u}), \boldsymbol{n}(\tilde{\boldsymbol{v}})) \\
& + \frac{1}{\left|\mathcal{V}_{\text{p}}^{\text{tgt}}\right|} \sum_{\boldsymbol{v} \in \mathcal{V}_{\text{p}}^{\text{tgt}}} 1 - \cos(\boldsymbol{n}(\boldsymbol{v}), \boldsymbol{n}(\tilde{\boldsymbol{u}})),
\end{aligned} \tag{8}$$

with $\tilde{\boldsymbol{v}}$ being the closest vertex in $\mathcal{V}_{\text{p}}^{\text{tgt}}$ to $\boldsymbol{u}$ and $\tilde{\boldsymbol{u}}$ being the closest vertex in $\mathcal{V}_{\text{gt}}^{\text{tgt}}$ to $\boldsymbol{v}$, i.e. $\tilde{\boldsymbol{v}} = \arg\min_{\mathbf{v} \in \mathcal{V}_{\text{p}}^{\text{tgt}}} \|\mathbf{u} - \mathbf{v}\|^2$ and $\tilde{\boldsymbol{u}} = \arg\min_{\mathbf{u} \in \mathcal{V}_{\text{gt}}^{\text{tgt}}} \|\mathbf{v} - \mathbf{u}\|^2$. $\boldsymbol{n}(\boldsymbol{v})$ is the normal vector for vertex $\boldsymbol{v}$, which is the averaged normal vectors of the faces that $\boldsymbol{v}$ belongs to.

### 6.1.4. Intra-mesh Normal Distance Loss

The normal distance loss that evaluates the consistency of adjacent face normal vectors in the predicted mesh is defined as $\mathcal{L}_{\text{norm,intra}}(\mathcal{M}_{\text{p}}^{\text{tgt}})$:

$$\mathcal{L}_{\text{norm,intra}}(\mathcal{M}_{\text{p}}^{\text{tgt}}) = \frac{1}{\left|\mathcal{E}_{\text{p}}^{\text{tgt}}\right|} \sum_{\substack{f_1 \cap f_2 = e \\ e \in \mathcal{E}_{\text{p}}^{\text{tgt}}}} 1 - \cos(\boldsymbol{n}(f_1), \boldsymbol{n}(f_2)), \tag{9}$$

where faces $f_1$, $f_2$ shares the edge $e$ and have the face normal vectors $\boldsymbol{n}(f_1), \boldsymbol{n}(f_2)$ respectively.

### 6.1.5. Edge Length Loss

$\mathcal{L}_{\text{edge}}(\mathcal{M}_{\text{p}}^{\text{tgt}})$ calculates the average of the edge lengths in the predicted mesh:

$$\mathcal{L}_{\text{edge}}(\mathcal{M}_{\text{p}}^{\text{tgt}}) = \frac{1}{\left|\mathcal{E}_{\text{p}}^{\text{tgt}}\right|} \sum_{(\boldsymbol{v}_1, \boldsymbol{v}_2) = e \in \mathcal{E}_{\text{p}}^{\text{tgt}}} \|\boldsymbol{v}_1 - \boldsymbol{v}_2\|^2 \tag{10}$$

### 6.1.6. Displacement Regularisation Loss

$\mathcal{L}_{\text{disp}}(\mathcal{D}_{\text{p}}, \mathcal{M}_{\text{p}}^{\text{tgt}})$ is defined as the sum of the derivative of displacements to constrain the predicted transformation to be smooth (Rueckert et al., 1999):

$$\mathcal{L}_{\text{disp}}(\mathcal{D}_{\text{p}}, \mathcal{M}_{\text{p}}^{\text{tgt}}) = \frac{1}{\left|\mathcal{V}_{\text{p}}^{\text{tgt}}\right|} \sum_{\boldsymbol{v} \in \mathcal{V}_{\text{p}}^{\text{tgt}}} \|\boldsymbol{d}_{\text{p}}(\boldsymbol{v}) - \sum_{\boldsymbol{v}_{\text{nbr}} \in \mathcal{N}(\boldsymbol{v})} w(\boldsymbol{v}, \boldsymbol{v}_{\text{nbr}}) \boldsymbol{d}_{\text{p}}(\boldsymbol{v}_{\text{nbr}})\|^2, \tag{11}$$

$$w(\boldsymbol{v}, \boldsymbol{v}_{\text{nbr}}) = \frac{1}{\|\boldsymbol{v} - \boldsymbol{v}_{\text{nbr}}\|} \left( \sum_{\boldsymbol{v}' \in \mathcal{N}(\boldsymbol{v})} \frac{1}{\|\boldsymbol{v} - \boldsymbol{v}'\|} \right)^{-1}. \tag{12}$$

For each vertex $\boldsymbol{v}$, the difference between the displacement $\boldsymbol{d}_{\text{p}}(\boldsymbol{v})$ and the averaged displacements of its neighbors which are weighted by the inverse of the edge length is computed as the displacement derivative.

## 6.2. Additional experimental results

An example of the manually labeled sub-regions is shown in Figure 4. All vertices of the mesh were divided into 10 categories based on 9 boundary planes selected manually.

The choice of loss used for training or implementing baseline models may affect the registration results. We experimented with some potential loss combinations on the reference to target registration and pairwise image registration tasks. The results are shown in Table 4 and Table 5, where the model is named as the network name-loss combination.

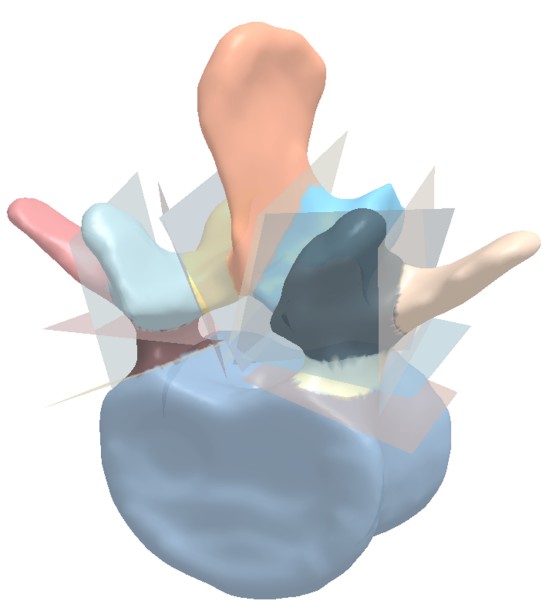

Figure 4: An example of the manually labeled sub-regions.

Table 4: TRE results in Reference to target registration.

| | SP | RL | LL | RTP | LTP |
|---|---|---|---|---|---|
| Ours | 2.11±1.09 | 2.13±1.31 | 1.66±0.97 | **2.01±1.39** | 2.34±1.93 |
| WeaklySup-BCE | 2.47±1.79 | 2.30±1.43 | 2.04±0.98 | 4.70±3.79 | 3.85±2.82 |
| WeaklySup-Dice | 3.06±2.01 | 2.43±1.40 | 2.51±1.16 | 4.41±2.68 | 5.15±2.37 |
| WeaklySup-BCE-SSD | 3.16±2.00 | 2.75±1.79 | 2.34±1.25 | 4.59±2.64 | 5.34±2.41 |
| Voxelmorph-BCE-SSD | 2.48±2.48 | 2.73±1.91 | 2.22±1.43 | 4.28±3.85 | 4.67±3.03 |
| Voxelmorph-Dice | 2.74±1.89 | 2.60±1.95 | 2.00±1.08 | 3.67±3.65 | 4.42±2.96 |
| Voxelmorph-Dice-SSD | 2.98±2.43 | 2.89±1.83 | 2.91±1.61 | 4.82±3.95 | 4.26±2.37 |
| Voxelmorph-Dice-NCC | 2.83±2.20 | 2.32±1.79 | 2.74±1.50 | 4.88±4.45 | 3.84±2.65 |
| NiftyReg-SSD | 4.03±2.94 | 3.44±2.52 | 4.36±3.57 | 6.62±4.98 | 8.51±4.58 |
| NiftyReg-NMI | 4.09±3.98 | 3.04±2.92 | 2.63±2.54 | 4.52±6.01 | 3.30±3.33 |

| | RAP | LAP | RP | LP | VB |
|---|---|---|---|---|---|
| Ours | 1.87±0.98 | 2.08±0.92 | 2.63±1.57 | 1.81±0.92 | 2.88±1.45 |
| WeaklySup-BCE | 2.46±1.88 | 2.03±1.04 | 3.74±2.62 | 2.43±1.43 | 2.31±1.70 |
| WeaklySup-Dice | 2.48±1.69 | 2.46±1.31 | 2.96±1.72 | 2.40±1.42 | 2.55±1.31 |
| WeaklySup-BCE-SSD | 2.39±1.31 | 2.50±1.30 | 2.65±1.44 | 2.58±1.48 | 2.94±1.74 |
| Voxelmorph-BCE-SSD | 2.37±1.97 | 2.16±1.14 | 3.84±2.54 | 3.13±1.45 | 2.88±1.98 |
| Voxelmorph-Dice | 2.52±2.17 | 2.30±1.31 | 4.35±3.13 | 3.81±1.79 | 2.85±2.09 |
| Voxelmorph-Dice-SSD | 2.79±2.49 | 2.51±1.95 | 5.02±2.79 | 4.07±2.18 | 3.96±2.82 |
| Voxelmorph-Dice-NCC | 3.15±2.38 | 2.72±1.36 | 4.80±3.35 | 4.05±1.96 | 3.05±1.92 |
| NiftyReg-SSD | 4.60±3.49 | 4.24±3.30 | 4.05±3.69 | 3.58±3.66 | 5.70±4.85 |
| NiftyReg-NMI | 2.82±3.19 | 1.86±1.45 | 3.56±3.90 | 2.45±1.98 | 7.57±4.80 |

Table 5: TRE results in Pairwise image registration.

|  | SP | RL | LL | RTP | LTP |
|---|---|---|---|---|---|
| Ours | 2.96±1.64 | 2.43±1.11 | 2.57±1.27 | **2.64±1.14** | **2.83±1.56** |
| WeaklySup-BCE | 4.06±1.88 | 3.97±2.14 | 4.31±2.24 | 6.30±3.99 | 6.07±3.13 |
| WeaklySup-Dice | 3.51±1.63 | 2.70±1.16 | 2.82±1.56 | 5.44±2.82 | 5.10±2.14 |
| WeaklySup-BCE-SSD | 3.72±1.65 | 3.58±1.52 | 3.12±1.42 | 5.50±3.29 | 5.32±2.37 |
| Voxelmorph-BCE-SSD | 4.39±2.07 | 4.07±2.04 | 3.92±1.93 | 6.67±3.91 | 6.10±3.17 |
| Voxelmorph-Dice | 5.60±2.35 | 5.50±2.48 | 5.15±2.58 | 7.08±4.33 | 8.72±4.31 |
| Voxelmorph-Dice-SSD | 5.75±2.53 | 5.97±2.70 | 5.39±2.88 | 8.54±5.27 | 7.38±4.34 |
| Voxelmorph-Dice-NCC | 5.23±2.39 | 5.33±2.83 | 4.74±2.23 | 7.45±4.41 | 7.32±3.77 |
| NiftyReg-SSD | 8.93±4.66 | 9.60±4.49 | 8.70±4.08 | 10.22±4.57 | 9.76±4.44 |
| NiftyReg-NMI | 10.39±5.07 | 11.89±6.42 | 10.28±5.02 | 14.38±5.72 | 13.23±5.16 |
|  | RAP | LAP | RP | LP | VB |
| Ours | **2.18±0.84** | **2.19±1.12** | 2.91±1.58 | **2.32±1.09** | 3.76±1.94 |
| WeaklySup-BCE | 3.01±1.79 | 3.74±1.89 | 4.28±2.76 | 3.49±1.85 | 4.43±2.21 |
| WeaklySup-Dice | 2.87±1.44 | 3.06±1.25 | 3.12±1.60 | 3.08±1.56 | 4.91±2.17 |
| WeaklySup-BCE-SSD | 3.04±1.46 | 3.00±1.35 | 3.32±1.59 | 3.10±1.57 | 4.34±2.20 |
| Voxelmorph-BCE-SSD | 4.28±1.89 | 4.12±1.79 | 5.02±2.81 | 3.81±2.25 | 5.49±2.39 |
| Voxelmorph-Dice | 5.79±2.70 | 5.20±2.20 | 6.64±3.37 | 5.57±2.63 | 6.75±3.08 |
| Voxelmorph-Dice-SSD | 6.23±3.84 | 5.44±2.06 | 6.33±3.34 | 5.54±2.81 | 8.27±4.17 |
| Voxelmorph-Dice-NCC | 5.34±2.50 | 5.31±2.20 | 5.76±3.19 | 4.96±2.78 | 6.55±3.04 |
| NiftyReg-SSD | 9.45±4.48 | 9.76±4.45 | 9.63±4.08 | 9.17±4.13 | 8.24±3.98 |
| NiftyReg-NMI | 11.44±5.28 | 11.68±5.25 | 11.89±5.37 | 11.15±4.69 | 10.33±5.50 |

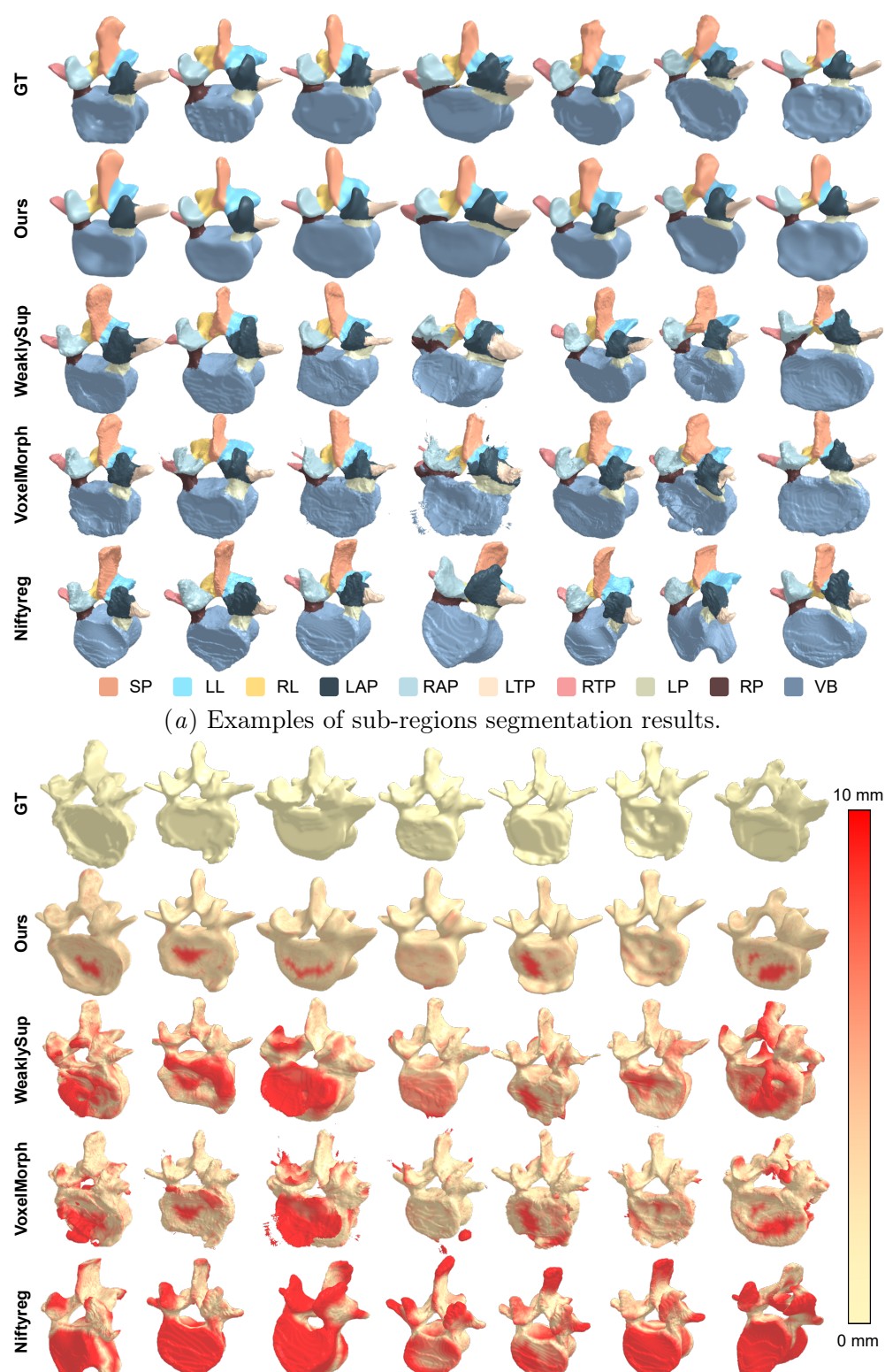

(a) Examples of sub-regions segmentation results.

(b) Example of vertebrae segmentation. The surface distance error is illustrated with red color.

Figure 5: Examples of vertebrae and sub-region segmentation results.

