# OpenReview forum: "Spatial Correspondence between Graph Neural Network-Segmented Images"
_MIDL.io/2023/Conference — MIDL 2023 Poster_

### Official Review · Reviewer_P45C · 2023-02-02

**Confidence:** 5
**Preliminary Rating:** 3
**Recommendation:** Poster

**Summary:**

This work employs an interesting combination of a GNN and a CNN for a mesh registration-based vertebra segmentation task. The (previously published) approach samples the features of a segmentation U-net along the normal of mesh vertices to inform the GNN about local image appearance (in a coarse-to-fine manner, first passing in the low-resolution features from the U-net's bottom). The combined model can be employed for registration by mapping the reference mesh used during training individually to both the target and reference image and then combining the transforms. The method is compared against two other publicly available registration algorithms, a SSD-based one from NiftyReg and a DL-based WeaklySup implementation. Three small public datasets are pooled into a training set of 24 subjects (about 120 vertebrae) and 11 test subjects, no validation set was used.

**Strengths:**

The approach uses an interesting, non-trivial setup and loss function and achieves good results. It should not be seen as a segmentation method, but its strength is that a reference mesh is placed onto the target volume, providing potentially useful correspondences between different subjects or timepoints.

The figures nicely illustrate the approach and the results of all compared methods.

**Weaknesses:**

I see two main issues with this work: One that can be much more easily be addressed during the revision process is the current presentation of the work; the readability is reduced by some language issues and imprecise descriptions, sometimes using different words for the same thing or without clear definition. The paper structure also does not yet very clearly discriminate between prior work or even just a general introduction of the registration process and own contributions.

The second issue I see is harder to address: Although the results look good, the small dataset size and the fact that no validation set was used make the evaluation results less trustworthy. I am not familiar enough with WeaklySup (or even its specific reimplementation used for comparison here) to say if this is a strong competitor, but I think NiftyReg could probably have been tuned to give better results. What's more, I find the statement "without using a validation set for hyperparameter tuning, which may further improve the performance" (which I assume to mean to say that there could have been improvements with hyperparameter tuning, even though it currently sounds as if the boost came from not using a validation set) very interesting: I can believe that no extensive hyperparameter tuning was performed, but it is hard to believe that all of the (many!) parameters that this method has were initially set and only one experiment was ever run / no experiment was run after looking at the results on the test set. In fact, the paper explicitly mentions "preliminary experiments", so the reader is left with some doubt that the authors could really refrain from ever testing their preliminary models on non-training cases.

**Deanonymize Review:**

yes

**Detailed Comments:**

It would be relevant to state the environment the meshing (marching cubes, Laplacian smoothing) was performed in.

The formula for the dynamic loss weighting contains a suspiciously placed division by π; I believe that should have been outside the parenthesis where it would be expected for normalization.

"based on which vertices were divided into 10 categories" neither specifies how many vertices were assigned categories (obviously all, if the colors in the images show the resulting sub-regions, but I first thought only the closest one, or a few local neighbors), nor how they were classified (geodesic voronoi?).

Similarly the "center of each category" is not well-defined, but leaves too much room for interpretation.

Should "9 planes" have been "9 faces"?  Or what's a "plane"?

The long list of references (Worchel et al., 2022; Munkberg et al., 2022; Wickramasinghe et al., 2020; Kong and Shadden, 2021; Kong et al., 2021; Bongratz et al., 2022) is unnecessarily repeated.

Figure 1 is very nice, but contains a few subfigures that should be labeled (e.g. a-d). Currently, the reader is referred to the figure for generic registration illustrations and will first look at the architectures and feature extraction, which is really unrelated (and could even have been an entirely separate figure).

The sentence "Particularly, for vertex v_i, denote..." interrupts the reading flow; you should reveal that the goal is still to explain the CNN feature extraction.

The "Variable ref." experiment is a robustness test, not an ablation study.

In the very generic explanation of pairwise registration process (not in the context of DL), you may not want to use the term "predicted" yet.

I suggest not to reference Figure 4 but just the appendix (which only has the figure anyhow).

Examples of English problems:
- are refereed to as
- More generally, pairwise registration method registers ...
- While, localising
- mixed data (-> "pooled"?)
- manually were selected
- accuracy, over the existing registration
- was applied for 30 times on
- regualrised
- were performed on Nvidia GPU
- significantly outperformed over
- the performance advantage over the alternative registration shared across ...
- orthopaedic

**Paper Type:**

both

**Questions To Address In The Rebuttal:**

I would like the authors to comment on the missing validation set (cf. my comments under "Weaknesses").

Did you also try NiftyReg with the (default) NMI measure, and could you comment on that?

The language issues of the paper should be addressed, and the precision of the presentation increased (see detailed comments).  There's not much to be discussed w.r.t. in the rebuttal, though.

---

### Official Review · Reviewer_C89N · 2023-02-03

**Confidence:** 3
**Preliminary Rating:** 3

**Summary:**

The authors propose a method to perform image segmentation alongside mesh registration using a mixture between graph and standard convolution networks.
The authors claim that GNN-based segmentation methods help in establishing dense correspondence between segmented ROIs, which in turn is useful for registration tasks.
The method is tested on several vertebrae datasets (LumSeg, SpiSeg, and xVertSeg).

**Strengths:**

Unfortunately, I could not fully understand the proposed methodology. Partly due to my limited intellectual capacity, but also due to flaws in the manuscript's narrative.
The method though shows quantitative improvements compared to existing methods (not sure though if these methods are SOTA).

**Weaknesses:**

I had difficulty reading the paper (many language and grammar mistakes) and understanding the methodology proposed by the authors (on top of the language issues, there are clear narrative flaws). Please see the more detailed comments I provide below.



**Deanonymize Review:**

no

**Detailed Comments:**

In arbitrary order:
1. I do not see how the discussion in the section on " Registration with a Reference Mesh" is connected to the method that you propose. Especially the pairwise image registration.
You have Eq. 4 which describes multiple losses from the Bongratz et al. paper. How the loss function and pairwise registration are connected?

2. You say "performs graph convolution with vertex features extracted from CNN module". From GNN I understand how to extract vertex features, but how do you extract vertex features from CNN?

3. "f_i_CNN is calculated by concatenating the sampled CNN embeddings H at multiple points along n" What is the sampled CNN embedding? What is voxel feature sampling in Fig. 1? Where does it enter the training? How the sampling is done?

4. What does define the selection of CNN image features that are sampled into GNN? It seems random {4567}, {3478}, ... And I still need to understand this voxel feature sampling.

5. How the segmentation and registration networks are trained? In parallel? If so, how do I get the meshes for the registration network? I assume that meshes are computed by marching cube algorithm from the segmentations, i.e. outputs of the segmentation network? So one needs to have two meshes for the registration network, but one gets them only after a forward pass through the segmentation net?

6. "Dp = fθ(Itgt,Mref)" is it really a function of both image and mesh, and not only mesh?

Please run through Grammarly or similar software. In the introduction section, grammar/language mistakes are in every other sentence.
- "Graph neural networks (GNNs) provides" no s
- " whilst GNNs have also been employed to represent non-grid prediction output for segmentation" make a new sentence
- "are refereed"
- "This work focus"
- "previously using convolutional neural networks" make a new sentence
- ...

It gets better in the method section, but still:
- "for reference mesh vertices from the image and reference mesh" (and reference mesh EDGES?)
- ...

**Paper Type:**

methodological development

**Questions To Address In The Rebuttal:**

Please address the points above and consider rewriting and restructuring the manuscript (e.g. you used the whole page 3 introducing mesh registration, while I am missing an important description in section 3 on how the pairwise registration or voxel sampling are used). I can believe that the proposed methodology is a breakthrough, but I was not able to get it, mainly due to issues with manuscript quality.

---

### Official Review · Reviewer_bvE6 · 2023-02-05

**Confidence:** 4
**Preliminary Rating:** 3

**Summary:**

Submission 90 develops a method for atlas-based image registration.

Methodologically, it makes use of images and their segmentation labels to train a hybrid image+mesh CNN+GNN which jointly trains for segmentation and alignment of a pre-specified mesh to the target image’s mesh. It develops a new regularization and a dynamic loss weighting scheme for it’s problem, with the rest of its formulation, various subnetworks, and losses largely derived from Vox2Cortex (Bongratz, et al. CVPR22).

Experimentally, it considers registration of spinal CT data. It benchmarks registration performance against two baselines (NiftyReg, which only uses intensities, and a deep learning framework which uses both intensities and segmentations) and presents a few ablations of its various subcomponents.


**Strengths:**

- The idea of using both intensity and semantic features with joint image and graph convolutional networks for image registration is well appreciated as combining both sets of features often leads to markedly improved registration performance.
- It is nice to see the atlas-based registration framework of Vox2Cortex extended also to pairwise image registration.
- Assuming that the reader is familiar with Vox2Cortex, the presentation of the paper is easy to follow and it has a direct message.


**Weaknesses:**

# Major
## Unclear contributions and differences w.r.t. Vox2Cortex?

Please correct me if I have misunderstood central contributions, but it appears that the submission largely directly applies the atlas-mesh deformation framework of Vox2Cortex (Bongratz, et al., cvpr22) which already does image registration with hybrid CNN+GNNs. In my reading, while the submission does appear to propose some loss term schedulers, it is quite unclear what exactly is the distinction between Vox2Cortex and this work as the submission’s title/abstract/introduction all build up a seemingly distinct framework but the methods largely derive from previous work.

In the rebuttal and/or revision, please clearly disambiguate the differences between the submission and the previous work that this paper draws from.

## Suboptimal baseline choices:
The submission presents experiments against two baselines: NiftyReg and WeaklySup. In my opinion, this is insufficient to actually demonstrate the benefits of the proposed method as:
- From an iterative optimization perspective:
    * NiftyReg only uses intensities and does not make use of shape information as in the proposed work. Semantic supervision (via labels/meshes/keypoints/etc) is known to greatly improve image registration performance and the submission should also compare with baselines that use both image+semantic information.
    * Further, NiftyReg is stated to be used with default hyperparameters without modification. As those hyperparameters were developed for very different data, their use for spinal CT unfairly hampers its performance.
- From a deep learning perspective:
    * Hybrid CNN/GNN-based registration with medical images and meshes has been exactly tackled in previous work such as [Sun, et al, MICCAI 21](https://miccai2021.org/openaccess/paperlinks/2021/09/01/449-Paper1738.html). While it is not necessary to compare against methods if they tackle extremely different tasks and domains, this missing reference should be at least discussed (and ideally benchmarked against, if possible).
    * While I am not strongly familiar with it, WeaklySup appears to be a reasonable inclusion as a comparable baseline that relies only on labels. However, the DL registration community has been extremely active in the time since it’s publication in 2018, and alternatives that use both intensities and labels such as LapIRN (Mok & Chung, MICCAI20 and its conditional variant at MICCAI21) or VoxelMorph with a hybrid NCC/Dice loss are likely better representatives of current work.
    * As this submission builds on Vox2Cortex, it can better distinguish itself by comparing directly to its exact formulation as an ablation, if possible. (please correct me if I missed this and it was included)

## Other experimental concerns:
- Statistical significance is claimed in most experiments. However, these experiments don’t appear to correct for multiple comparisons with the Bonferroni correction or equivalent and hence need to be revisited and corrected.
- I found the ablations to be difficult to parse as they seem to be unstructured and partially adhoc choices. The ablations would greatly benefit from being restructured in presentation to isolate individual distinctions from Vox2Cortex.
- Suboptimal train/test/val split: There does not seem to be a validation set used at all (section 4.1) and the test data is said to be held-out. Training deep networks without a validation set is likely suboptimal and, to clarify, was only the train set used for all model prototyping and development with the test left until the very end?


**Deanonymize Review:**

no

**Detailed Comments:**

# Minor:
- While Vox2Cortex is mentioned often throughout the text, concurrent work at MIDL2022 (TopoFit, Hoopes et al) proposed very similar work and should be mentioned in some capacity, in my opinion.
- Notational improvement: page 3, \mathcal{M}_{p}^{tgt} is confusing and would be clearer if it were to be \mathcal{M}_{p}^{moved} instead.
- There are a few typos that should be easy to catch with an automated spell-checker in a revision. E.g., the title for section 5 is misspelled.
- Section 1, the paragraph that starts with “One solution could be..”: this paragraph can be removed to make space for more important information as it is ultimately dismissed by the paper as “out-of-scope” anyway.


**Paper Type:**

validation/application paper

**Questions To Address In The Rebuttal:**

My most critical concerns revolve around:
- Clarification of distinctions w.r.t. previous work (Vox2Cortex in particular). Is this work different? If yes, please elaborate on the differences.
- Please better justify the choice of the two baselines in this submission (niftyreg and weaklysup) and address the points made in the relevant parts of the review above.

Of course, if space permits, rebuttals to all points are welcome. If I misunderstood key components, please let me know and I would be happy to reconsider my score.

# EDIT AFTER REBUTTAL:
I thank the authors for the discussion. 2/3 of my major concerns were somewhat resolved by the rebuttal (baselines and experimental concerns).

I am raising my score to a 'borderline' score as I am still not adequately convinced that this is a distinct contribution from the work it is building on (as that was also indirectly a registration paper) and it does not compare against other hybrid CNN+GNN registration methods that have been published recently ([example](https://miccai2021.org/openaccess/paperlinks/2021/09/01/449-Paper1738.html)).

---

### Official Review · Reviewer_dWXc · 2023-02-05

**Confidence:** 4
**Preliminary Rating:** 4
**Recommendation:** Oral

**Summary:**

The submission introduces a template-based GNN segmentation tool that can also establish spatial alignment robustly. Much of the inspiration / framework has come from Bongratz 2022 for both CNN and GNN work. The application is the segmentation of spinal vertebrae from CT images. The experiments are run on a decent size datasets with promising results and low target registration results compared to baseline methods

**Strengths:**

The submission is well written and mostly clear throughout the text.

Novelty: obtaining local spatial correspondence from new template-based GNN segmentation solution.

I like how the ablation study from the Appendix is summarized briefly but with all key information included.

Nice Figure 41

Concise clear conclusion which does not over claim or overstate the meaning of the experimental results and ablation study outcomes.

**Weaknesses:**

The submission should be read through carefully one more time to correct grammatical errors and spelling mistakes as well as find very long sentences.

It would be great if it was explicitly stated what the adaption from Bongratz2022 was and what is the novel part


**Deanonymize Review:**

no

**Detailed Comments:**

Some of the discussion when describing methodology is very focused on vertebrae, when I tough it could be more general.

Fig 2: There is no green. And actually there are three colors. Update caption!

I could not parse the last sentence of 4.4, and unfortunately did not get the discussion point.

**Paper Type:**

both

**Questions To Address In The Rebuttal:**

Can provide a few sentences about how/why you decided to select your baseline tools for the performance comparison?

The paper very much focused on the vertebrae segmentation/registration problem. What other anatomies / modalities would best benefit from this technique?

---

### Meta-Review · Area_Chair_bLtL · 2023-02-22

**Recommendation:** Accept (Poster)
**Confidence:** 5

**Metareview:**

Spatial Correspondence between Graph Neural Network-Segmented Images

The consensus is on categorizing this submission as a borderline paper. The contribution is on applying a hybrid mesh-image, GNN and CNN for registration by aligning template structures. The main concern is a close similarity of concepts with vox2cortex [Bongratz'22], followed by partly addressed validation issues. While the contribution indeed sounds borderline with limited novelty, the recommendation is leaning towards acceptance given the global ranking of submissions.

Recommendation towards Acceptance.